# Effect of intra-articular dexmedetomidine on experimental osteoarthritis in rats

**Lyvia Maria Rodrigues de Sousa Gomes**[1,2☯], **Nicolau Gregori Czeczko**[3☯], **Rayanne Luiza Tajra Mualem Araújo**[1‡], **Maria do Socorro de Sousa Cartagenes**[1,2‡], **José Osvaldo Barbosa Neto**[2☯]*, **João Batista Santos Garcia**[1,2☯]

**1** Department of Anesthesia, Pain and Palliative Care, Federal University of Maranhão, Maranhão, Brazil,
**2** Experimental Laboratory for the Study of Pain, Federal University of Maranhão, Maranhão, Brazil,
**3** Evangelic Faculty of Paraná, Paraná, Brazil

☯ These authors contributed equally to this work.
‡ These authors also contributed equally to this work
* osvaldo1983@me.com

**Data Availability Statement:** Data is available in a public repository. DOI: 10.6084/m9.figshare.12482762.

## Abstract

Pharmacological treatment of osteoarthritis is still inadequate due to the low efficacy of the drugs used. Dexmedetomidine via the intra-articular (i.a.) route might be an option for the treatment of osteoarthritis-associated pain. The present study assessed the analgesic and anti-inflammatory effects of dexmedetomidine administered via the i.a. route in different doses in an experimental model of rat knee osteoarthritis induced with monosodium iodoacetate. Rats were allocated to four groups with 24 animals in each group. The OA (osteoarthritis), DEX-1 (dexmedetomidine in dose of 1μg/kg) and DEX-3 (dexmedetomidine in dose of 3μg/kg) groups were subjected to induction of osteoarthritis through injection of monosodium iodoacetate (MIA) via the i.a. route on the right knee; the control group was not subjected to osteoarthritis induction. Clinical assessment was performed on day 0 (before osteoarthritis induction) and then on days 5, 10, 14, 21 and 28 after induction. Treatment was performed on day 7 via the i.a. route, consisting of dexmedetomidine in doses of 1 and 3 μg/kg, while group OA received 0.9% normal saline. The animals were euthanized on days 7, 14, 21 and 28. Samples of the synovial membrane were collected for histopathological analysis, and the popliteal lymph nodes were collected for measurement of cytokines (interleukin [IL] IL-6, tumor necrosis factor alpha [TNF-α]). Dexmedetomidine (1 and 3 μg/kg) significantly reduced the animals' weight distribution deficit during the chronic-degenerative stage of osteoarthritis and improved the pain threshold throughout the entire experiment. Histological analysis showed that dexmedetomidine did not cause any additional damage to the synovial membrane. The TNF-α levels decreased significantly in the DEX-3 group on day 28 compared with the OA group. Dexmedetomidine reduced pain, as evidenced by clinical parameters of osteoarthritis in rats, but did not have an anti-inflammatory effect on histological evaluation.

**Funding:** The author(s) received no specific funding for this work.

**Competing interests:** The authors have declared that no competing interests exist.

## Introduction

Osteoarthritis (OA) is a degenerative disease characterized by progressive degeneration of the knee cartilage accompanied by secondary inflammation of the synovial membrane. This disease is defined as a heterogeneous group of conditions that cause painful joint signs and symptoms associated with defects of joint cartilage integrity [1].

Cytokines stand out among the pro-inflammatory mediators involved; their levels are increased in arthritic joints in both humans and animals. Cytokines play fundamental roles in the development of pain and inflammation, especially tumor necrosis factor alpha (TNF-α) and interleukin IL-6 [2].

Pharmacological treatment of OA is still inadequate due to the low analgesic efficacy and adverse effects of the drugs currently used. Clonidine [3], fadolmidine [4] and dexmedetomidine [5,6] have been studied to improve the current understanding of their local and systemic analgesic effects, so that they might be better used for treatment of diseases involving inflammation and pain, such as OA.

Dexmedetomidine is a super-selective α2-adrenergic agonist with considerable sedative and analgesic actions. The mechanism of its intra-articular analgesic action is not yet fully elucidated, but it is similar to clonidine. Clonidine acts on presynaptic receptors, inhibiting the release of norepinephrine in peripheral afferent receptors, and exhibits local anesthetic action via inhibition of stimuli conduction through C and A-delta fibers [7]. In addition, clonidine via the intra-articular route reduces the pain behavior in experiments with animals and promotes postoperative analgesia when used alone or in combination with bupivacaine [8].

There are more studies on intra-articular dexmedetomidine use in acute pain than in chronic diseases, such as OA [9,10]. Nevertheless, some studies have shown that the analgesia achieved through this route is better compared with other routes of administration and minimized effects such as hypotension and bradycardia [7,11].

As dexmedetomidine has been little investigated, particularly when administered via the intra-articular route, and it exhibits similarities to other α2 agonists, such as clonidine, the objective of the present study was to test the hypothesis that intra-articular dexmedetomidine has analgesic and anti-inflammatory actions on monosodium iodoacetate-induced knee OA in rats.

## Methods

### Study design and experimental groups

A total of 96 adult male Wistar rats weighing 180 to 200 g and 60 days of age were used following approval by the research ethics committee of the Federal University of Maranhão, Brazil.

The animals were acclimatized over a period of eight days under noise control conditions, temperature of 22°C ± 2°C and relative humidity of 40% to 60%, with a 12-hour light/dark cycle. The animals were given rat feed (Purina®, São Paulo, Brazil) and water *ad libitum*. Each animal was placed on the glass chamber and on the Von Frey's test platform for 5 minutes the day before the experiments, and once again on the day of the experiment, to allow them to be familiarized with the devices used.

The animals were randomly allocated into four groups (A, B, C and D), each group containing 24 rats. Each of these groups were then subdivided in smaller groups (n = 6) to be euthanized on the four specific study times (D7, D14, D21 e D28) for tissue harvest. On day 1, OA was induced in groups B, C and D using monosodium iodoacetate (MIA). Group A was the control group and was not subjected to OA induction. Group B, designated as the OA group, was treated with saline solution. Groups C and D were treated with dexmedetomidine via the

intra-articular route, the difference being the drug dose (1 and 3 μg/kg, respectively). In all the groups, the volume of treatment was 50 μl. The dosages used in this study were based on previous publications with dexmedetomidine but opting for the second dose to be slightly higher (3μcg/kg) than those referenced, with the objective of evaluating a possible analgesic and anti-inflammatory effect of the drug [7,12].

### Osteoarthritis induction

On day zero, the animals were weighed using an electronic scale and anesthetized using 40 mg/kg of sodium thiopental (Cristália, São Paulo, Brazil) via the intraperitoneal route. This procedure was repeated on the days when treatments were applied.

OA was induced through a single intra-articular injection of 2 mg of MIA diluted to a maximum volume of 50 μl and applied to the right knee (using a syringe and disposable sterile insulin needle [BD®, 26G X 3/8]). For the injection, the limb was flexed at the knee level to approximately 90˚, and the MIA solution was injected into the intra-articular space through the patellar ligament between the tibia and the femur [13–15]. On day 7 after induction, the following procedures were performed according to the group allocation:

- Control: observation;

- OA: injection of 50 μl of 0.9% normal saline;

- DEX-1: injection of 50 μl of solution containing 1 μg/kg of dexmedetomidine to the right knee;

- DEX-2: 50 μl of solution containing 3 μg/kg of dexmedetomidine to the right knee.

### Clinical assessment

Clinical assessment for behavioral signs of pain was performed before induction of OA (day zero) and then at regular intervals. Five days after intra-articular administration of MIA, the animals were subjected to clinical assessment, corresponding to assessment day 5. Seven days after OA induction, groups B, C and D were administered the corresponding treatments as described above, and clinical assessments for behavioral signs of pain were performed on days 0, 5, 10, 14, 21 and 28 (Fig 1). The early stage of OA was defined as the period until day 14 after induction, and the chronic-degenerative stage of OA was defined as the period starting on day 21 [16]. Clinical assessment was performed using weight-bearing and Von Frey's devices.

### Joint incapacitation (weight-bearing test)

The degree of joint incapacitation was assessed based on the changes in the weight distribution on the hind paw between the right and left limbs, which reflects the degree of discomfort of the affected joint in the animals with OA [17,18].

The animals were placed in an angled glass chamber so that each hind paw rested on a separate platform. A period of 5 minutes was allowed for adaptation before assessment; measurements were taken once the animals were in the correct position.

The weight bear on each hind paw (in grams) was determined over a 5-second period; the mean of three measurements was considered for analysis [17,18].

### Mechanical allodynia (Von Frey's test)

Used to assess the paw withdrawal threshold, the Von Frey's device employs a digital an algesimeter (Insight, São Paulo, Brazil) and consists of a pressure transducer connected to a digital

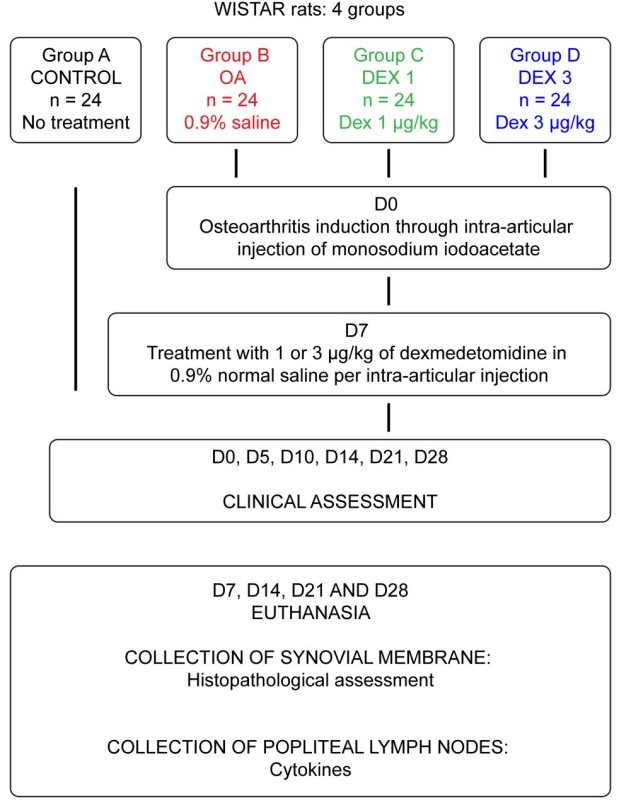

**Fig 1. Flowchart representing the experimental protocol.**

strength meter (in grams). The device is calibrated to achieve a maximal force of 150 g, with 0.1-gram precision until a force of 80 g. The pressure transducer contacts the animal's paw through a duly adapted 0.5-mm diameter disposable polypropylene tip. The same investigator applied linearly increasing pressure on the center of the paw until the animal exhibited a response characterized as flinching of the stimulated paw. The stimuli were repeated up to six times until three similar measurements were achieved with clear flinching after paw withdrawal [18].

## Euthanasia

The animals from all groups were euthanized with 150 mg of sodium thiopental via the intra-peritoneal route. Death was defined as respiratory arrest and complete absence of reflexes [19]. Animals from all four groups were anaesthetized and euthanized on days 7, 14, 21 and 28; on day 7, euthanasia was performed 6 hours after treatment for OA. Following euthanasia, the popliteal lymph nodes and the synovial membrane of each animal's right knee were removed for histopathological analysis.

## Sample collection and study of cytokines in the popliteal lymph nodes

After popliteal lymph node extraction, the tissue was placed in a numbered petri dish containing sterile Roswell Park Memorial Institute (RPMI) medium. Then, the material was taken to the flow chamber to be macerated. After that, cells suspensions were dyed using Crystal Violet solution. The cells were then quantified in a Neubauer chamber using a light optical microscope, and cell adjustment was performed ($10^6$ cells/mL). The cells were placed in flat-

bottomed 48 wells culture plates and incubated in $CO_2$ incubator for 48h. After this period, the cell culture supernatant was collected and stored at -80˚C for subsequent cytokine dosage [20]. Lymph nodes were harvest on days 7, 14, 21 and 28 of the study. IL6 and TNF-α were measured using the ELISA method. Cytokine analysis was performed by a biochemist specializing in immunophysiology who was blinded to the control and experimental groups.

## Histopathological analysis of the synovial membrane

Histological analysis of the synovial membrane was performed by a pathologist who was blinded to the control and experimental groups. In all groups, histopathological analysis was performed, following identical protocol, on days 7,14,21 and 28 (Fig 1). The synovial membrane was extracted and fixed in 10% buffered formaldehyde. After 48 hours, the sample was processed through a standardized procedure until its inclusion in the paraffin blocks. The compartments were longitudinally separated and sent for routine histological preparation by hematoxylin-eosin. The tissue sections were placed in rectangular molds, forming blocks that were subsequently sectioned by 4 μm microtome steel knives [21,22].

The following parameters were interpreted: presence of EDEMA, based on the presence of inflammatory exudate and impairment of inflammatory exudate; presence of INFILTRATE, based on analysis of the polymorphonuclear infiltrate; and presence of FIBROSIS, based on the presence of fibroid deposits. These parameters were selected based on findings from studies that used experimental models of OA in Wistar rats [21,23]. In the analysis of the histological sections, a score was attributed to each analyzed parameter (Table 1).

## Statistical analysis

Sample size was calculated using G*Power software [25]. The number of animals was calculated using ANOVA test to obtain a 35% difference between groups for the Von Frey's test, with 80% power and 0.05 alfa error, obtaining a total of 96 animals, divided by four groups (n = 24). For the histological study, 6 animals of each group were used for tissue harvesting at days 7, 14, 21 e 28. The data were analyzed using IBM SPSS Statistics 20, version 2011. Normal distribution of the data was tested by Shapiro Wilk's test. Since data did not follow a normal distribution, Kruskal Wallis test was performed to access statistical difference for histological and comportamental variables between groups. Whenever an effect was deemed significant, Dunn's test was used to perform 2 x 2 comparisons. Thus, analyses were performed to assess the group effects (control, OA, DEX1 and DEX-3) on days 7, 14, 21 and 28. For the weight bearing and Von Frey's test variables, the assessment time-points were days 0, 5, 10, 14, 21 and 28. The significance level to test the null hypothesis was 5%, i.e., the results were considered to be significant when $p < 0.05$.

## Results

### Joint incapacitation test

Following injection of MIA, all the induced animals exhibited signs of joint discomfort, predominantly bearing weight on the healthy paw, when compared with the controls.

**Table 1. Parameter scores [24].**

| Score 0 | Absence of parameter |
|---|---|
| Score 1 | Parameter present at mild intensity |
| Score 2 | Parameter present at moderate intensity |
| Score 3 | Parameter present at strong intensity |

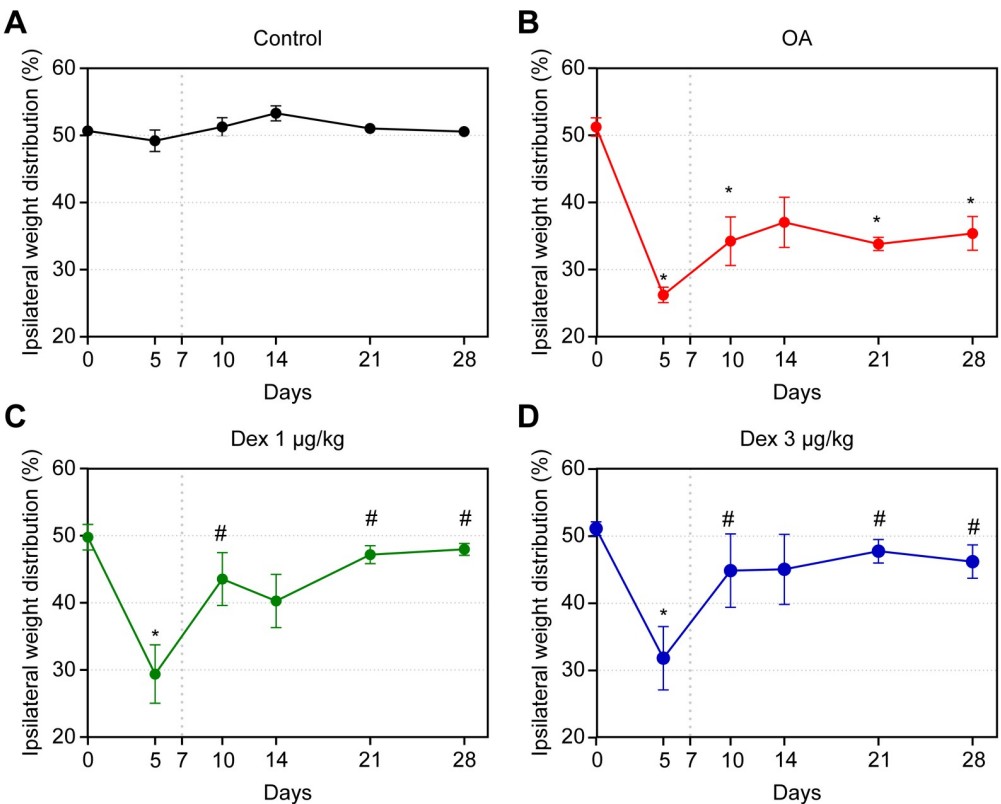

**Fig 2. Assessment of weight distribution on the hind paws in rats treated with normal saline (0.9% NaCl, i.a.) or dexmedetomidine at doses of 1 μg/kg or 3 μg/kg, i.a.** Symbols and vertical lines represent the mean ± standard error of the mean. The vertical dotted lines represent the onset of treatment. * indicates a significant difference compared with the control group. # indicates a significant difference compared with the OA group.

In the animals with OA, the changes induced by MIA followed a biphasic distribution. In the first phase, there was a marked reduction in the weight placed on the affected paw; this effect reached its maximum on day 5 after induction, indicating that the difference in weight distribution on the paws was significant between the control group and all others ($p < 0.05$). On day 10, significant differences were noted between the OA group and the DEX-1 and DEX-3 groups ($p < 0.05$), with the results of the latter two groups being close to those of the control. The second phase began on day 14 after induction; there was a reduction in the weight placed on the affected paw in the OA group compared with the control. This limitation remained until the end of the experiment, with significant differences on days 21 and 28 ($p < 0.05$).

In the DEX-1 and DEX-3 groups, improvements in the weight distribution deficit between the hind paws were observed on days 21 and 28; this effect was statistically significant compared with the OA group ($p < 0.05$). No significant difference was observed between the groups treated with different doses of dexmedetomidine (Fig 2).

## Mechanical allodynia (Von Frey's test)

On day 5, the OA, DEX-1 and DEX-3 groups exhibited significant reductions in pain threshold compared with the control group ($p < 0.05$). After the onset of treatment, the pain thresholds improved in the DEX-1 and DEX-3 groups, compared with the OA group, and were

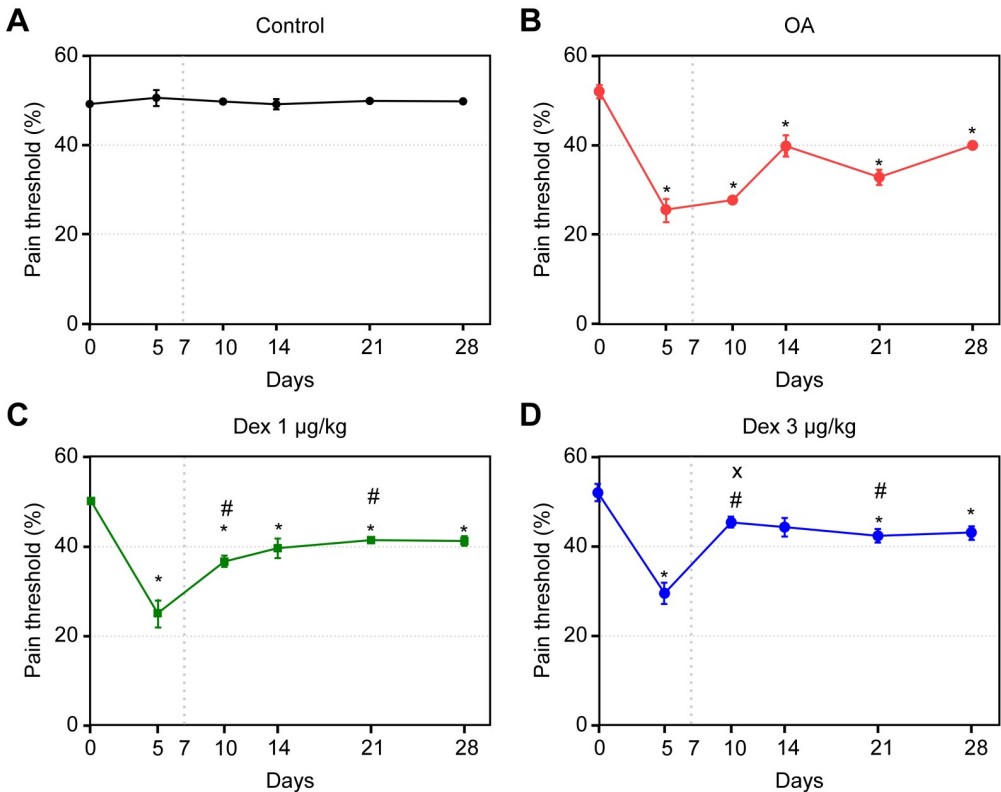

**Fig 3. Assessment of tactile allodynia in rats treated with normal saline (0.9% NaCl, i.a.) or dexmedetomidine (1 μg/kg or 3 μg/kg, i.a.).** Symbols and vertical lines represent the mean ± standard error of the mean. The vertical dotted line represents the onset of treatment. * indicates a significant difference compared with the control group. # indicates a significant difference compared with the OA group. + indicates a significant difference compared with the DEX 3 μg/kg group.

maintained in both the acute and chronic-degenerative stages; these differences were significant on days 10 and 21 ($p < 0.05$). On day 10, the values corresponding to the DEX-3 group, compared with those of the DEX-1 group, were closer to those of the controls; this difference was significant ($p < 0.05$) (Fig 3).

## Histopathological analysis of the synovial membrane

Intra-articular injection of MIA caused histological changes in the synovial membranes of the animals with induced OA; the differences between the control group and all others were significant for all parameters assessed ($p < 0.05$). None of the assessed parameters differed significantly among the OA, DEX-1 and DEX-3 groups. Notably, dexmedetomidine did not cause additional histological damage to the membrane, as the manifestations of edema, cell infiltrate and fibrosis were the same as those exhibited by the group treated with normal saline (Table 2 and Fig 4).

## Assessment of IL-6

The IL-6 levels increased significantly ($p < 0.05$) in the OA group compared with the control group on day 7. In the later stages of OA, there were no significant differences between these groups.

**Table 2. Histopathological analysis of the synovial membrane of the control, OA, DEX-1 and DEX-3 groups relative to the parameter edema, cell infiltrate and fibrosis.**

| EDEMA | | | | | |
|---|---|---|---|---|---|
| **GROUP** | **DAY** | | | | **p value** |
| | **7** | **14** | **21** | **28** | |
| **Control** | 0.00 ± 0.00 | 0.00 ± 0.00 | 0.00 ± 0.00 | 0.00 ± 0.00 | 0.1250 |
| **OA** | 1.33 ± 0.21 | 1.66 ± 0.21 | 1.83 ± 0.40 | 1.00 ± 0.00 | 0.1250 |
| **DEX 1** | 1.66 ± 0.21 | 1.33 ± 0.21 | 1.00 ± 0.00 | 2.00 ± 2.44 | 0.1250 |
| **DEX 3** | 1.50 ± 0.22 | 2.50 ± 0.50 | 0.83 ± 0.16 | 0.66 ± 0.21 | 0.1250 |
| CELL INFILTRATE | | | | | |
| **GROUP** | **DAY** | | | | |
| | **7** | **14** | **21** | **28** | |
| **Control** | 0.00 ± 0.00 | 0.00 ± 0.00 | 0.00 ± 0.00 | 0.00 ± 0.00 | |
| **OA** | 1.50 ± 0.22 | 1.66 ± 0.42 | 2.00 ± 0.44 | 1.00 ± 0.00 | 0.1250 |
| **DEX 1** | 2.16 ± 0.16 | 2.00 ± 0.44 | 1.00 ± 0.00 | 2.00 ± 0.44 | 0.0975 |
| **DEX 3** | 2.00 ± 0.00 | 1.50 ± 0.50 | 1.33 ± 0.33 | 0.66 ± 0.21 | 0.0975 |
| FIBROSIS | | | | | |
| **GROUP** | **DAY** | | | | |
| | **7** | **14** | **21** | **28** | |
| **Control** | 0.00 ± 0.00 | 0.00 ± 0.00 | 0.00 ± 0.00 | 0.00 ± 0.00 | |
| **OA** | 1.83 ± 0.30 | 1.66 ± 0.21 | 2.00 ± 0.36 | 2.60 ± 0.24 | 0.1250 |
| **DEX 1** | 2.33 ± 0.21 | 1.83 ± 0.30 | 2.00 ± 0.31 | 2.00 ± 0.36 | 0.0975 |
| **DEX 3** | 1.83 ± 0.40 | 2.25 ± 0.75 | 2.33 ± 0.33 | 1.33 ± 0.49 | 0.1250 |

NOTE: Statistically significant difference was tested with Kruskal-Wallis test.

Administration of dexmedetomidine (1 or 3 µg/kg) did not significantly reduce the IL-6 levels at any time-point of assessment. However, the IL-6 levels of the DEX-3 group exhibited a trend towards reduction, coming closer to those of the control group at all time-points of assessment compared to the OA group (Table 3).

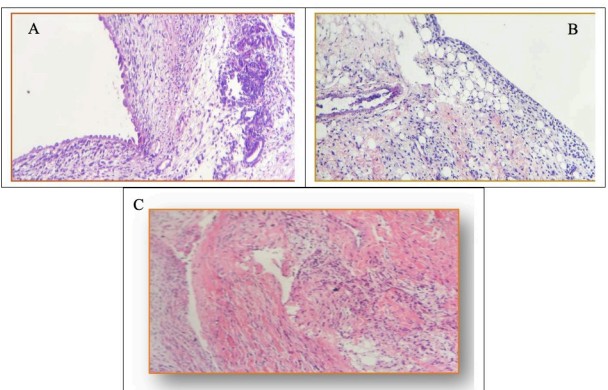

**Fig 4. Histopathological analysis of the synovial membrane.** A) Edema and moderate chronic lymphoplasmacytic inflammatory infiltrate with formation of lymphoid follicle and synovial hyperplasia. B) Edema and moderate chronic lymphoplasmacytic inflammatory infiltrate with focal areas of fibrosis. C) Inflammatory infiltrate with intense fibrosis of the synovial membrane.

**Table 3. Assessment of IL-6 levels in lymph node supernatants by means of the Kruskal-Wallis and Dunn's tests.**

| VARIABLE | GROUP | DAY | | | | | | | |
|---|---|---|---|---|---|---|---|---|---|
| | | 7 | p | 14 | p | 21 | p | 28 | p |
| IL-6 (lymph node) | Control | 0.138 [b] | 0.003 | 0.124 | 0.840 | 0.117 | 0.084 | 0.105 | 0.232 |
| | Osteoarthritis | 0.190 [a] | | 0.124 | | 0.124 | | 0.110 | |
| | DEX-1 | 0.191 [a] | | 0.120 | | 0.115 | | 0.126 | |
| | DEX-3 | 0.168 [ab] | | 0.118 | | 0.106 | | 0.132 | |

NOTE

[a, b] Different letters represent $p < 0.05$ by Dunn's test.

## Assessment of TNF-α

The TNF-α levels were significantly higher in the OA group than in the control group on days 7, 14 and 28.

Although the levels of TNF-α decreased in the DEX-1 and DEX-3 groups compared with the OA group at all time-points of assessment (days 7, 14, 21 and 28), the differences were not significant except on day 28, when the TNF-α levels were significantly lower in the DEX-3 group than in the OA group (Table 4).

## Discussion

The effects of intra-articular MIA followed a biphasic distribution; the first stage lasted until the 14th day after induction, which marked the beginning of the second stage. MIA model caused inflammation inside the joint cavity, perceived by significant reduction in the gait score on day 5 after induction, when the animals began to exhibit signs of joint discomfort. Also, inflammatory histopathological abnormalities of the synovial membrane, such as edema, polymorphonuclear infiltrate and fibrosis, as documented in other studies [3,14]. These findings are compatible with other experimental models of OA in rats [3,26] and might be explained by the rapid inflammatory effect of MIA on the articular cartilage, followed by a late chronic-degenerative stage, which includes a neuropathic component and the subchondral bone's abnormalities.

In the present study, dexmedetomidine was able to reduce joint discomfort in both the acute and chronic-degenerative stages of OA, as evidenced in the joint incapacitation test; the groups treated with dexmedetomidine showed improvement compared with the untreated group. These results agree with several studies reported in the literature that have demonstrated the antinociceptive effect of dexmedetomidine, and an adjuvant action that potentiates postoperative analgesia [5,12,27–30]. The improvements observed on days 21 and 28 after

**Table 4. Assessment of TNF-α levels in lymph node supernatants by means of the Kruskal-Wallis and Dunn's tests.**

| VARIABLE | GROUP | DAY | | | | | | | |
|---|---|---|---|---|---|---|---|---|---|
| | | 7 | P | 14 | p | 21 | p | 28 | p |
| TNF-α (lymph node) | Control | 0.135 [b] | 0.041 | 0.151 [b] | 0.022 | 0.166 | 0.710 | 0.157 [b] | 0.020 |
| | Osteoarthritis | 0.158 [a] | | 0.180 [a] | | 0.180 | | 0.207 [a] | |
| | DEX-1 | 0.144 [ab] | | 0.165 [ab] | | 0.178 | | 0.172 [a] | |
| | DEX-3 | 0.144 [ab] | | 0.182 [a] | | 0.184 | | 0.158 [b] | |

NOTE

[a, b] Different letters represent $p < 0.05$ by Dunn's test.

induction of OA in the animals treated with dexmedetomidine indicate that the drug also acts in the chronic-degenerative stage of the disease, with similar effects at doses of 1 or 3 μg/kg. These findings agree with the results of an experimental study that used a model of formalin-induced pain in rats, which demonstrated the antinociceptive action of dexmedetomidine in both the acute (phase 1) and later (phase 2) phases of hyperalgesia [30].

Induction of OA with MIA caused a considerable reduction in the pain threshold, and treatment with dexmedetomidine at both tested doses improved this threshold up to the end of the study, exhibiting dose-dependence on day 10. One experimental study used two models of OA induction in rat knees, MIA and meniscectomy, with consequent development of allodynia and hyperalgesia. Morphine was used as treatment, and there were significant reductions in the pain thresholds in both models, indicating that morphine administered via the subcutaneous route is an efficacious treatment [14]. Several studies showed that hyperalgesia is mediated by α-2 adrenergic receptors and that use of adrenergic antagonists decreases the pain threshold [6].

The effect of dexmedetomidine in the chronic stage of OA was detected in both clinical tests; this effect might be related to both the central and peripheral actions of the drug. The mechanism underlying the intra-articular analgesia caused by dexmedetomidine has not yet been clearly defined; however, it might be similar to the mechanism suggested for clonidine, an α-2 agonist that acts on presynaptic α-2 adrenergic receptors, inhibiting the release of norepinephrine in peripheral afferent nociceptors [31].

The antinociceptive effects of α-2 agonists are not only limited to their central actions. Several studies showed that intra-articular dexmedetomidine reinforces postoperative analgesia after arthroscopic knee surgery [7]. Dexmedetomidine and clonidine exhibited a local anesthetic effect in different studies [32–34]. One study that used fadolmidine, a highly selective α-2 agonist with limited central action, administered via the i.a. route to rat knees with induced OA reported suppression of the pain-related behavior due to selective actions on the peripheral α-2 adrenergic and opioid receptors [4].

Animals with chronic joint injury, as in the model used in the present study, may show signs and symptoms with neuropathic characteristics due to abnormalities of nociceptive processing. Neuropathic pain encompasses a broad range of pain behaviors, and some studies have shown that it is associated with OA in both humans and experimental models [13,35–38]. One experimental study conducted with rats that assessed the systemic use of tramadol combined with dexmedetomidine for treatment of neuropathic pain found that the tested combination in low doses was effective in increasing the pain threshold [35]. Future studies with animal models of neuropathic pain using dexmedetomidine administration via the local route should be performed to establish whether this drug might be a therapeutic target for treatment of the neuropathic components of chronic pain in OA.

Analysis of the synovial membrane revealed significant inflammatory responses in the affected joints in animals subjected to MIA-induced OA, coinciding with the early stage of joint discomfort, which might have been caused by inflammatory infiltrates in the synovial membrane. Other studies also detected an inflammatory response characterized by edema, cell infiltrate and fibrosis after experimental MIA-induced OA [15,17,39]. Our results show that dexmedetomidine did not reduce edema, cell infiltrate or fibrosis, suggesting that, in the doses used, the drug did not have a consistent local anti-inflammatory action.

No additional histological damage to the membrane due to the use of dexmedetomidine was detected, which suggests that the i.a. administration of this drug is safe. Several studies gathered evidence on the analgesic action of dexmedetomidine administered via the i.a. route in humans [7,40]; however, none investigated the possible presence of histological damage caused by the drug. One study assessed anti-inflammatory agents, such as naproxen, rofecoxib

and acetaminophen, administered via the oral route to rats with MIA-induced OA, and histo-pathological analysis did not detect any alteration in the articular cartilage [14,39].

The increase in the levels of TNF-α and IL-6 in OA [1,41], similar to the ones found in the present study (especially on days 7 and 10 after OA induction) corroborate to the inflammation mediated by the injected MIA. The effect of the i.a. injection of MIA on day 7, which represents the most acute stage of OA, resulted in increases in pro-inflammatory cytokines. These increases were followed by the chronic-degenerative stage, starting on day 14 after induction, when the levels of these cytokines decreased. If the lymph nodes had been collected before day 7, the expression levels of acute-phase cytokines may have been even higher than observed. Other anti-inflammatory cytokines that participate in the chronic stage of OA, and thus with higher expression at that stage, such as IL-1, IL-4, IL-10, IL-11, IL-13 and interferon (IFN)-γ, could also have been assessed [41,42].

In the present study, following treatment with dexmedetomidine, the two analyzed cyto-kines–TNF-α and IL-6 –exhibited trends towards reduction. These pro-inflammatory mole-cules, i.e., TNF-α and IL-6, are known to be some of the critical mediators of inflammation within the context of the pathophysiology of OA [6]. These results suggest that dexmedetomi-dine can mitigate inflammation after i.a. administration, although the reduction in cytokines did not reach statistical significance in our study. In a study that used a different model for OA, more consistent reductions in inflammatory cytokines were observed [6]. However, the mechanism behind this anti-inflammatory effect has yet to be elucidated.

Recently, the roll of the inhibition of Nod-like receptor pyrin domain 3 (NLRP3) inflamma-some by dexmedetomidine was investigated. This receptor is considered a molecular switch and is responsible for activation of caspase-1 and, finally, by the upregulation of IL-1β and Il-18, known pro-inflammatory cytokines. The effectiveness of the inhibition of the activation of NLRP3 in the improvement of OA progression was already demonstrated in an animal model [6]. Down-regulation of inflammatory factors were obtained by the inhibition of nuclear fac-tor-κB (NF-κB) pathway with dexmedetomidine, through α-2A adrenergic receptors subtype in septic rats. The activation of NF-κB induces the transcription of proinflammatory cytokines such as TNF-α and IL-6 [6,43].

Considering the data from a previously mentioned research [6], combined with the findings of the present study, suggest a dose-dependent effect for the anti-inflammatory action of i.a dexmedetomidine. The doses studied in both experiments were 1, 3, 5, 10 and 20mcg/kg, and significantly reduction in cytokines was only obtained with 10 and 20mcg/kg. Future studies ought to investigate this phenomenon.

One study that employed a model of inflammation induced by local injection of carra-geenin in the hindpaws of mice showed that high doses of dexmedetomidine administered via local injection significantly inhibited TNF-α and cyclooxygenase (COX)-2 expression, which suggests an anti-inflammatory effect mediated by α-2 adrenoreceptors [44]. In a study that assessed the analgesic and immunomodulatory effects of dexmedetomidine in a model of for-malin-induced inflammatory pain, the TNF-α and IL-6 levels in the plasma and spleen lym-phocyte supernatant decreased, although the differences were not significant [30].

One limitation of the present study is that it was not possible to precisely define the anti-inflammatory role of dexmedetomidine or to define the mechanism that might have caused the various changes in the levels of the three assessed cytokines. Therefore, future experimental studies using different doses of dexmedetomidine, collecting other materials (e.g., articular cartilage) and assessing other mediators or other cytokines should be performed to help iden-tify an effective treatment against pain and inflammation progression in OA.

This study shows that dexmedetomidine produced an improvement in pain behavior in OA rats demonstrated by a reduction in joint incapacitation on days 10, 21 and 28, and

mechanical alodinia after day 10 and 21. In the histological evaluation of the synovial membrane, dexmedetomidine did not produce an anti-inflammatory effect.

## Author Contributions

**Conceptualization:** Lyvia Maria Rodrigues de Sousa Gomes, Rayanne Luiza Tajra Mualem Araújo, João Batista Santos Garcia.

**Data curation:** Lyvia Maria Rodrigues de Sousa Gomes, Rayanne Luiza Tajra Mualem Araújo.

**Formal analysis:** Lyvia Maria Rodrigues de Sousa Gomes.

**Methodology:** Lyvia Maria Rodrigues de Sousa Gomes.

**Project administration:** Rayanne Luiza Tajra Mualem Araújo, Maria do Socorro de Sousa Cartagenes, João Batista Santos Garcia.

**Resources:** Maria do Socorro de Sousa Cartagenes.

**Supervision:** Nicolau Gregori Czeczko, Maria do Socorro de Sousa Cartagenes, José Osvaldo Barbosa Neto, João Batista Santos Garcia.

**Validation:** Nicolau Gregori Czeczko, Maria do Socorro de Sousa Cartagenes, João Batista Santos Garcia.

**Writing – original draft:** Lyvia Maria Rodrigues de Sousa Gomes.

**Writing – review & editing:** Nicolau Gregori Czeczko, José Osvaldo Barbosa Neto, João Batista Santos Garcia.

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
