## [Decision Letter · Decision Letter 0]

1 Jun 2020

PONE-D-20-10457

Effect of intra-articular dexmedetomidine on experimental osteoarthritis in rats

PLOS ONE

Dear Dr. José Osvaldo Barbosa Neto,

Thank you for submitting your manuscript to PLOS ONE. After careful consideration, we feel that it has merit but does not fully meet PLOS ONE’s publication criteria as it currently stands. Therefore, we invite you to submit a revised version of the manuscript that addresses the points raised during the review process.

Dear Authors,

the reviewer found several major critiques to your manuscript.Please try to address her/his points.

We look forward to receiving your revised manuscript.

Kind regards,

Francesco Staffieri

Academic Editor

PLOS ONE

Journal Requirements:

Additional Editor Comments (if provided):

Dear Authors,

the reviewer found several major critiques to your manuscript.Please try to address her/his points.

Reviewers' comments:

Reviewer's Responses to Questions

**Comments to the Author**

1. Is the manuscript technically sound, and do the data support the conclusions?

Reviewer #1: Yes

2. Has the statistical analysis been performed appropriately and rigorously? 

Reviewer #1: No

3. Have the authors made all data underlying the findings in their manuscript fully available?

Reviewer #1: No

4. Is the manuscript presented in an intelligible fashion and written in standard English?

Reviewer #1: No

5. Review Comments to the Author

Reviewer #1: 1 - In the experimental design there is no control group that has received only the intra-articular saline solution, this is an invasive operation which in itself can create tissue damage, it would be better to compare the effects with this group.

Furthermore, it would be useful to compare intra-articular dexmedetomidine to another intra-articular reference drug. Please, justify why a control drug was not used to compare the two responses

2 - The Authors should explain to this reviewer how therapeutic doses were chosen

3- Line 145-146

"During this period, the animals were also acquainted with the devices that were later used to assess the presence of behavioural signs of pain".

The authors should specify how this occurred/was performed, for how long per day the animals familiarized themselves with the devices.

4- Line 149

“24 rats each after four weeks and were clustered into groups of six animals per group”.

This sentence is unclear, please reformulate.

5 - Line 234

“The lymph nodes were collected on days 7, 14, 21 and 28”

Please specify how many animals have been sacrificed each time? and why did the Authors choose to keep samples on a numbered plate containing bacteriostatic medium for in vitro culture and were they kept in the refrigerator at -10 ° C? why was this storage temperature chosen and not -20 ° C or -80 ° C? should Authors specify how long the samples were kept and how they were processed for subsequent analysis?

6 - Did the Authors notice differences in popliteal lymph nodes sizes? why did the authors not also consider the number of cells present in the samples?

7 “Histopathological analysis of the synovial membrane “

how was this analysis performed and on which samples? on which section? how were the samples processed? what coloring was made? furthermore, the authors should show the images of this analysis for each group.

8 Statistical analysis

Line 261-262,

To this reviewer is not clear which statistical test was performed by the authors. Please, clarify.

Moreaover, Authors should clarify the sample size for each analysis and how it was chosen.

9 In the discussion, the Authors mention conclusions that are not supported by their results, but are based on literature. A specific statement should be set on what is the novelty and the benefits that this study can give.

6. PLOS authors have the option to publish the peer review history of their article (what does this mean?). If published, this will include your full peer review and any attached files.

Reviewer #1: No

---

## [Author Response · Author response to Decision Letter 0]

25 Jun 2020

Reviewers' comments:

Comments to the Author

1. Is the manuscript technically sound, and do the data support the conclusions?

Reviewer #1: Yes

No Response.

2. Has the statistical analysis been performed appropriately and rigorously? 

Reviewer #1: No

Response: All data were rigorously collected and then analyzed through proper statistical tests using IBM SPSS Statistics 20, version 2011.

3. Have the authors made all data underlying the findings in their manuscript fully available?

The PLOS Data policy requires authors to make all data underlying the findings described in their manuscript fully available without restriction, with rare exception (please refer to the Data Availability Statement in the manuscript PDF file). The data should be provided as part of the manuscript or its supporting information or deposited to a public repository. For example, in addition to summary statistics, the data points behind means, medians and variance measures should be available. If there are restrictions on publicly sharing data—e.g. participant privacy or use of data from a third party—those must be specified.

Reviewer #1: No

Response: All data was made available at a public repository.

DOI: 10.6084/m9.figshare.12482762

Link: https://figshare.com/s/935f9e049480fbd4f428

4. Is the manuscript presented in an intelligible fashion and written in standard English?

Reviewer #1: No

Response: All language problems were addressed and corrected.

Reviewer #1: 

1 - In the experimental design there is no control group that has received only the intra-articular saline solution, this is an invasive operation which in itself can create tissue damage, it would be better to compare the effects with this group.

Furthermore, it would be useful to compare intra-articular dexmedetomidine to another intra-articular reference drug. Please, justify why a control drug was not used to compare the two responses.

Response: We do not believe that the inclusion of a second control group that receives only saline solution would enrich the results obtained in this study. Although saline injection can produce some inflammation, it is expected to be acute. The model chosen for this study was that of a chronic inflammation induced by monosodium iodoacetate (MIA). The objective of the control group used was to demonstrate that the MIA in fact produced chronic articular alterations. Regarding the comparison of dexmedetomidine to other drugs, the purpose of our study was to establish if dexmedetomidine was at all effective in this scenario, and not yet to evaluate superiority over any other drugs. 

2 - The Authors should explain to this reviewer how therapeutic doses were chosen

Response: The dosages used in this study were based on previous publications with dexmedetomidine, but opting for a second dose slightly higher (3μcg/kg) than those referenced, with the objective of evaluating a possible analgesic and anti-inflammatory effect of the drug. AL-(METWALLI et al., 2008; VILELA E NASCIMENTO JÚNIOR, 2003 ; PIAO, G.E WU J. , 2014.) Clarification done in the manuscript.

3- Line 145-146

"During this period, the animals were also acquainted with the devices that were later used to assess the presence of behavioural signs of pain".

The authors should specify how this occurred/was performed, for how long per day the animals familiarized themselves with the devices.

Response: Each animal was placed on the glass chamber and on the von frey test platform for 5 minutes the day before the experiments, and once again on the day of the experiment, to allow them to be familiarized with the devices used. (Alteration done in the manuscript.)

4- Line 149

“24 rats each after four weeks and were clustered into groups of six animals per group”.

This sentence is unclear, please reformulate.

Response: The animals were randomly allocated into four groups (A, B, C and D), each group containing 24 rats. Each of these groups were then subdivided in smaller groups (n=6) to be euthanized on the four specific study times (D7, D14, D21 e D28) for tissue harvest. (Modification done on the manuscript)

5 - Line 234

“The lymph nodes were collected on days 7, 14, 21 and 28”

Please specify how many animals have been sacrificed each time? 

Response: The lymph nodes were collected on days 7, 14, 21 and 28. At each one of these days, 6 animals of each one of the groups were euthanized and the tissues were harvested. Alteration done in the manuscript.

Why did the Authors choose to keep samples on a numbered plate containing bacteriostatic medium for in vitro culture and were they kept in the refrigerator at -10 ° C? 

Response: This method is the standard procedure to processing tissue for cytokine study, as developed by Moore, Gerner e Franklin (1967). The same protocol was previously used in Calado GP, Lopes AJO, Costa Junior LM, Lima FdCA, Silva LA, Pereira WS, et al. (2015) Chenopodium ambrosioides L. Reduces Synovial Inflammation and Pain in Experimental Osteoarthritis. PLoS ONE 10(11): e0141886. https://doi.org/10.1371/journal.pone.0141886. Reference now included in the manuscript

Why was this storage temperature chosen and not -20 ° C or -80 ° C? 

Response: The refrigerator used was in fact a -80ºC, not the -10ºC, as wrongly stated on the manuscript (correction done on the manuscript).

Should Authors specify how long the samples were kept and how they were processed for subsequent analysis?

Response: After popliteal lymph node extraction, the tissue was placed in a numbered petri dish containing sterile Roswell Park Memorial Institute (RPMI) medium. Then, the material was taken to the flow chamber to be macerated. After that, cells suspensions were dyed using Crystal Violet solution. The cells were then quantified in a Neubauer chamber using a light optical microscope, and cell adjustment was performed (106 células/ml). The cells were placed in flat-bottomed 48 wells culture plates and incubated in CO2 incubator for 48h. After this period, the cell culture supernatant was collected and stored at -80°C for subsequent cytokine dosage. Lymph nodes were harvest on days 7, 14, 21 and 28 of the study. (Clarification added to the manuscript)

6 - Did the Authors notice differences in popliteal lymph nodes sizes? why did the authors not also consider the number of cells present in the samples?

Response: The comparison between lymph node size or cell count was not part of our study. Therefore, only cell adjustment was performed, following the technical standards for immunohistochemistry.

7 “Histopathological analysis of the synovial membrane “

how was this analysis performed and on which samples? on which section? how were the samples processed? what coloring was made? furthermore, the authors should show the images of this analysis for each group.

Response: In all groups, histopathological analysis was performed, following identical protocol, on days 7,14,21 and 28 (figure 1 with flowchart). The synovial membrane was extracted and fixed in 10% buffered formaldehyde. After 48 hours, the processing was followed by routine methods until the inclusion in paraffin blocks. The compartments were longitudinally separated and sent for routine histological preparation by hematoxylin-eosin. The tissue sections were placed in rectangular molds, forming blocks that were subsequently sectioned by 4 μm microtome steel knives (CARDOSO, 2007; CASTRO, 2008).

*8.Statisticaanalysis

Line 261-262

**To this reviewer is not clear which statistical test was performed by the authors. Please,clarify.

Response: Normal distribution of the data was tested by Shapiro Wilk’s test. Since data did not follow a normal distribution, Kruskal Wallis test was performed to access statistical difference for histological and comportamental variables between groups. Whenever an effect was deemed significant, Dunn’s test was used to perform 2 x 2 comparisons. (Clarification done in the manuscript).

**Moreaover, Authors should clarify the sample size for each analysis and how it waschosen.

Response: Sample size was calculated using G*Power software [17]. The number of animals was calculated using ANOVA test to obtain a 35% difference between groups for the von frey test, with 80% power and 0.05 alfa error, obtaining a total of 96 animals, divided by four groups (n=24). For the histological study, 6 animals of each group were used for tissue harvesting at days 7, 14, 21 e 28. (Alteration made on the manuscript).

*9. In the discussion, the Authors mention conclusions that are not supported by their results, but are based on literature. A specific statement should be set on what is the novelty and the benefits that this study can give.

Response: This study shows that dexmedetomidine produced an improvement in pain behavior in OA rats demonstrated by a reduction in joint incapacitation on days 10, 21 and 28, and mechanical alodinia after day 10 and 21. In the histological evaluation of the synovial membrane, dexmedetomidine did not produce an anti-inflammatory effect, nor did it damage to the synovial membrane, suggesting that its intra-articular administration is safe. (Alteration done in the manuscript)

---

## [Decision Letter · Decision Letter 1]

22 Sep 2020

PONE-D-20-10457R1

Effect of intra-articular dexmedetomidine on experimental osteoarthritis in rats

PLOS ONE

Dear Dr. José Osvaldo Barbosa Neto,

Thank you for submitting your manuscript to PLOS ONE. After careful consideration, we feel that it has merit but does not fully meet PLOS ONE’s publication criteria as it currently stands. Therefore, we invite you to submit a revised version of the manuscript that addresses the points raised during the review process.

Please try to satisfy the requests of the reviewer

We look forward to receiving your revised manuscript.

Kind regards,

Francesco Staffieri

Academic Editor

PLOS ONE

Additional Editor Comments (if provided):

Please try to satisfy the requests of the reviewer

Reviewers' comments:

Reviewer's Responses to Questions

**Comments to the Author**

1. If the authors have adequately addressed your comments raised in a previous round of review and you feel that this manuscript is now acceptable for publication, you may indicate that here to bypass the “Comments to the Author” section, enter your conflict of interest statement in the “Confidential to Editor” section, and submit your "Accept" recommendation.

Reviewer #1: (No Response)

2. Is the manuscript technically sound, and do the data support the conclusions?

Reviewer #1: Partly

3. Has the statistical analysis been performed appropriately and rigorously? 

Reviewer #1: Yes

4. Have the authors made all data underlying the findings in their manuscript fully available?

Reviewer #1: Yes

5. Is the manuscript presented in an intelligible fashion and written in standard English?

Reviewer #1: Yes

6. Review Comments to the Author

Reviewer #1: In the reviewer opinion, there remain some few questions that are attached:

- Authors should show representative images for histopathological analysis (hematoxylin and eosin);

- The Authors report that DEX produced an improvement in pain. As dexmedetomidine do not have anti-inflammatory effect, how do the Authors explain the effects of dexmedetomidine on pain? Please clarify.

- The Authors conclude that intra-articular administration of dexmedetomidine is safe. However the Authors did not show the histopathological results of the control groups (animals without OA) treated with intra-articular DEX-1 and DEX-2.

7. PLOS authors have the option to publish the peer review history of their article (what does this mean?). If published, this will include your full peer review and any attached files.

Reviewer #1: No

---

## [Author Response · Author response to Decision Letter 1]

30 Nov 2020

Response to reviewers:

The questions and suggestions made by the reviewers were addressed right after each of the items exposed in the document sent by the scientific journal.

Comments to the Author

1. If the authors have adequately addressed your comments raised in a previous round of review and you feel that this manuscript is now acceptable for publication, you may indicate that here to bypass the “Comments to the Author” section, enter your conflict of interest statement in the “Confidential to Editor” section, and submit your "Accept" recommendation.

Reviewer #1: (No Response)

2. Is the manuscript technically sound, and do the data support the conclusions?

Reviewer #1: Partly

3. Has the statistical analysis been performed appropriately and rigorously? 

Reviewer #1: Yes

4. Have the authors made all data underlying the findings in their manuscript fully available?

Reviewer #1: Yes

5. Is the manuscript presented in an intelligible fashion and written in standard English?

Reviewer #1: Yes

6. Review Comments to the Author

Reviewer #1: In the reviewer opinion, there remain some few questions that are attached:

- Authors should show representative images for histopathological analysis (hematoxylin and eosin);

RESPONSE: Histopathological images were included as suggested by the reviewer.

- The Authors report that DEX produced an improvement in pain. As dexmedetomidine do not have anti-inflammatory effect, how do the Authors explain the effects of dexmedetomidine on pain? Please clarify.

RESPONSE: Clarification for this item was already addressed in the DISCUSSION section.

The effect of dexmedetomidine in the chronic stage of OA was detected in both clinical tests; this effect might be related to both the central and peripheral actions of the drug. The mechanism underlying the intra-articular analgesia caused by dexmedetomidine has not yet been clearly defined; however, it might be similar to the mechanism suggested for clonidine, an α-2 agonist that acts on presynaptic α-2 adrenergic receptors, inhibiting the release of norepinephrine in peripheral afferent nociceptors [31]. 

- The Authors conclude that intra-articular administration of dexmedetomidine is safe. However, the Authors did not show the histopathological results of the control groups (animals without OA) treated with intra-articular DEX-1 and DEX-2.

RESPONSE: This statement was removed from the conclusion.

In the histological evaluation of the synovial membrane, dexmedetomidine did not produce an anti-inflammatory effect.

7. PLOS authors have the option to publish the peer review history of their article (what does this mean?). If published, this will include your full peer review and any attached files.

Do you want your identity to be public for this peer review? For information about this choice, including consent withdrawal, please see our Privacy Policy.

Reviewer #1: No

---

## [Decision Letter · Decision Letter 2]

28 Dec 2020

Effect of intra-articular dexmedetomidine on experimental osteoarthritis in rats

PONE-D-20-10457R2

Dear Dr. José Osvaldo Barbosa Neto,

We’re pleased to inform you that your manuscript has been judged scientifically suitable for publication and will be formally accepted for publication once it meets all outstanding technical requirements.

Kind regards,

Francesco Staffieri

Academic Editor

PLOS ONE

Additional Editor Comments (optional):

Reviewers' comments:

Reviewer's Responses to Questions

**Comments to the Author**

1. If the authors have adequately addressed your comments raised in a previous round of review and you feel that this manuscript is now acceptable for publication, you may indicate that here to bypass the “Comments to the Author” section, enter your conflict of interest statement in the “Confidential to Editor” section, and submit your "Accept" recommendation.

Reviewer #1: All comments have been addressed

2. Is the manuscript technically sound, and do the data support the conclusions?

Reviewer #1: Yes

3. Has the statistical analysis been performed appropriately and rigorously? 

Reviewer #1: Yes

4. Have the authors made all data underlying the findings in their manuscript fully available?

Reviewer #1: Yes

5. Is the manuscript presented in an intelligible fashion and written in standard English?

Reviewer #1: Yes

6. Review Comments to the Author

Reviewer #1: (No Response)

7. PLOS authors have the option to publish the peer review history of their article (what does this mean?). If published, this will include your full peer review and any attached files.

Reviewer #1: No

---

## [Editor Report · Acceptance letter]

4 Jan 2021

PONE-D-20-10457R2 

Effect of intra-articular dexmedetomidine on experimental osteoarthritis in rats. 

Dear Dr. Barbosa Neto:

I'm pleased to inform you that your manuscript has been deemed suitable for publication in PLOS ONE. Congratulations! Your manuscript is now with our production department. 

Kind regards, 

on behalf of

Dr. Francesco Staffieri 

Academic Editor

PLOS ONE